# Multispecialty Approach to a Very Large Congenital Head and Neck Cystic Lymphatic Malformation in an Infant Born by SARS-CoV-2 Positive Mother—A Case Report

**DOI:** 10.3390/biomedicines10102422

**Published:** 2022-09-28

**Authors:** Greta Sibrecht, Katarzyna Wróblewska-Seniuk, Jakub Kornacki, Daniel Boroń, Jarosław Szydłowski, Anna Kłosowska, Ewa Bień, Ewa Wender-Ożegowska, Tomasz Szczapa

**Affiliations:** 1II Department of Neonatology, Poznan University of Medical Sciences, 61-701 Poznan, Poland; 2Department of Reproduction, Poznan University of Medical Sciences, 61-701 Poznan, Poland; 3Department of Pediatric Otorhinolaryngology, Poznan University of Medical Sciences, 61-701 Poznan, Poland; 4Department of Pediatrics, Hematology and Oncology, Medical University, 80-210 Gdansk, Poland

**Keywords:** lymphatic malformation, congenital cervical lesion, EXIT procedure, sirolimus, neck malformation

## Abstract

Masses of the head and neck are often diagnosed prenatally and require special care due to the risk of airway obstruction. The EXIT procedure is a preferable mode of delivery. A congenital cystic lymphatic malformation is one of the most common lesions of the cervical region described in neonates. The treatment consists of different strategies and involves the cooperation of multiple specialists. Up to now, no guidelines or protocols are available. We report a case of a congenital cystic lymphatic malformation of the head and neck delivered during the EXIT procedure by a mother who was SARS-CoV-2 positive. We analyzed clinical characteristics, radiologic features, and treatment with injections of sclerotic agents and orally administrated sirolimus. Sirolimus seems a valuable and safe therapeutic option for treating lymphatic malformations, especially with adjunct therapies.

## 1. Introduction

Congenital head and neck cystic lesions are relatively uncommon. They are usually painless, soft, or fluctuant and might cause airway obstruction [1]. The most common types are thyroglossal duct cysts and cystic hygroma (lymphatic malformation) [2]. Four types of lymphatic malformation can occur in the head and neck region: capillary, cavernous, cystic, and venolymphatic. All types might also be seen within the same lesion during the histopathological examination. A cystic lymphatic malformation is the most common of these forms and is responsible for around 5% of benign lesions in infancy and childhood [3]. A lymphatic malformation is related to PIK3CA mutations, which can also be found in many types of cancer, such as breast, lung, brain, or gastric tumors [4].

Lymphatic malformations start to develop prenatally around the 6th week of gestation by the sequestration of embryonic lymphatic channels or insufficient drainage of the lymphatic sac into the jugular vein, which causes congenital obstruction of lymphatic drainage [5]. Approximately 5% of these lesions are present at birth [6]. The most common locations of cystic lymphatic malformations are the posterior part of the neck and the oral cavity. A cystic lymphatic malformation is a multilocular cystic mass with septa visible on the ultrasound. Sometimes hemorrhagic components can be visualized inside the cysts [7]. MRI has better resolution for soft tissues, so it is superior to CT to plan the therapy [8].

Any large mass within the head and neck can cause airway obstruction at birth. For this reason, the ex-utero intrapartum treatment (EXIT) procedure is a preferable way of delivery. It allows keeping the placental circulation by controlling uterine hypotonia while securing the fetal airways just before the delivery of the newborn [9,10].

The possible management of cystic lymphatic malformation after the delivery includes observation, aspiration, injection with sclerotherapy agents, treatment with the mammalian target of rapamycin (mTOR) inhibitor, and excision. However, there are neither algorithms nor guidelines for the neonatal population.

We present a case of a newborn born with massive head and neck cystic lymphatic malformation treated multidisciplinary with sclerotic agents, sirolimus, and surgery.

## 2. Case Report

The patient was a late preterm baby boy born at 34 weeks of gestation via a cesarean section. The diagnosis of midline fetal neck lesion was made for the first time at 29th week gestation, when the diameter of the lesion was 68 × 65 mm^2^ (Figure 1). The mother was referred to the tertiary center specializing in fetal neck malformations at 32 weeks of gestation. Two prior pregnancies were uncomplicated and delivered vaginally on time. On the day of hospital admission, the mass’s diameter was 85 × 59 mm^2^ (Figure 2). The lesion had mixed both cystic and solid echogenicity in ultrasound. Differential diagnoses included teratoma and lymphatic malformations. No other fetal abnormalities were found. However, moderate polyhydramnios was diagnosed (Amniotic Fluid Index of 30 cm). We managed to complete full steroid treatment for fetal lung maturation. Due to polyhydramnios and increasing uterine contractions despite intravenous tocolysis, the cesarean section with EXIT procedure had to be performed at 34th-week gestation. At the time of delivery, the mother was SARS-CoV-2 positive; therefore, all the procedures required extra protection. Due to this infection, we could not perform magnetic resonance imaging (MRI) before birth.

After partial head extraction, the baby was monitored immediately in the delivery room, and an otolaryngologist tried twice to intubate the newborn before umbilical cord clamping. The tube’s position in the trachea was confirmed with an endoscope after the first attempt at intubation. However, due to the lack of adequate ventilation and bradycardia, the umbilical cord was clamped and cut, and the newborn was transferred to the neonatal resuscitation table to be intubated by a neonatologist. The neonate weighed 2550 g, APGAR scores at 1 min were 1, 4 at 5 min, and 5 at 10 min.

On the physical examination, the malformation of the head and neck had dimensions of 21 cm (between two ear lobes) × 14 cm (horizontal) × 8 cm (vertical). No signs of dysmorphia or abnormalities of the chest, heart, abdomen, genitalia, and back were found. Figure 3 shows the baby’s lesion on the first day of life.

After admission to the neonatal intensive care unit, the baby required mechanical ventilation with room air and low ventilation parameters, possibly due to airway obstruction caused by the malformation. Serial tests for COVID-19 infection remained negative. He was sedated and initially fed parenterally. On the 6th day of life, the MRI examination was performed (see Figure 4), which revealed a heterogenous lesion of the head and neck extending from the craniofacial area, embracing the lower lip and the bottom of the oral cavity, the superior mediastinum, and the upper part of the chest on the right side. The lesion consisted of numerous cystic structures with the presence of septa and fluid levels visible.

During the first days of the infant’s life, the mass was enlarging, and the newborn’s head was compulsorily positioned in deflect position.

On the 7th day of life, the baby presented with fever and high C-reactive protein (CRP) level and was given antibiotics which were continued for three weeks. The blood culture was positive for *E. coli*. The cerebrospinal fluid culture was negative. The patient required red blood cell transfusions due to anemia and fresh frozen plasma transfusion due to abnormal coagulation values. Anemia and abnormal coagulation values were probably due to consumptive coagulopathy and bleeding into the cysts. He was also treated with furosemide and albumin due to edemas and hypoalbuminemia. Edemas were observed in the lesion area due to malfunctioning lymphatic vessels and very limited motor activity of the baby. Hypoalbuminemia was most probably reactive to edema. The patient was fed exclusively enterally from the 11th day of life via a nasogastric tube and later via gastrostomy.

On the 16th day of life, the patient underwent the first surgery that was supposed to reduce the mass. During the procedure, the malformation was punctured, lymphatic fluid was aspirated, and 3 mg of doxycycline was given as a percutaneous sclerotherapy procedure. Fiberoscopy performed on that day revealed massive edema of the tissues of the larynx and glottis. On the root of the tongue, the lymphangioma was more extensive than seen during the EXIT procedure. Surgical removal of the malformation was not possible at that point. It was expected that doxycycline would cause a reduction of the mass. On the 19th day of life, the patient underwent a tracheotomy. Obtaining adequate access to the trachea forced a partial resection of the mass. The parts limiting access to the anterior wall of the trachea were removed. Initially, the baby required mechanical ventilation with low parameters and afterward CPAP through the tracheostomy until the 78th day of life. After that, he was breathing spontaneously.

Recovery from the surgery was complicated by poor wound healing. We observed tissue necrosis and serous-bloody liquid leaking from the wound. Despite treatment with Bactigrass (paraffin dressing with chlorhexidine) and Spongostan (gelatin sponge), no improvement was seen, and a subsequent surgical intervention was performed to clean the wounds. Antibiotics were continued. Due to continuous enlargement of the malformation, on the 34th day of life, the baby required subsequent doxycycline injection in the dose of 3 mg into the mass.

On the 35th day of life, after obtaining the parents’ informed consent and the bioethical committee’s approval, we started the off-label treatment with an oral solution of sirolimus (Rapamune). The initial dose was 0.2 mg every 12 h, followed by an increase to 0.3 mg every 12 h based on the plasma levels (target range 4–8 µg/L) and the infant’s weight gain.

As far as Rapamune side effects are concerned, we observed transient leucopenia with the lowest white blood cell count 2.42 G/L. Within two weeks, the white blood cell count was back to normal. Cholesterol and triglyceride levels remained stable. The baby developed lymphatic head edema that was treated with lymphatic massage with slight improvement during the following weeks (Figure 5).

The mass showed a change in consistency after two weeks of treatment—it became more solid and stopped growing. Also, the tongue showed a visible reduction in size after sirolimus administration. On the ultrasound, we visualized hypoechogenic cysts (the biggest of 2 cm in diameter). On the 58th day of life, the malformation was punctured again (1 mL of fluid aspired), and OK432 (Picibanil) in the dose of 0.2 mg was injected. During this procedure, similarly to doxycycline injection, the obliterating preparation was administered to three malformation chambers with a diameter exceeding 20 mm. During this intervention, bronchofiberoscopy was performed, and no pathological changes were observed. A control MRI was performed on the 70th day of life. It showed an extensive multicystic lesion with multiple septa penetrating both sides of the oral cavity, parotid salivary gland, and sublingual space, between the muscles of the oral cavity floor, including the whole submandibular area (see Figure 6a,b). The mass also penetrated around sternocleidomastoid muscles, covering the visceral space of the neck anteriorly from the jugular vessels. In the area at the bottom of the mouth, the venous vessels were widened up to 5 mm. Several foci and hemosiderin deposits were visible within the lesion that might have corresponded to the bleeding within the cysts. The entire lesion’s size was similar to the previous MRI examination, while the cysts seemed smaller.

After an interdisciplinary case conference, the baby was disqualified from debulking surgery. The convened council decided that the patient requires additional pharmacological therapy to reduce the mass further. On the 90th day of life, the patient was transported to the Pediatric Oncology Department to receive optimally adjusted dosing and continue treatment with sirolimus. Currently, the baby is seven months old and remains on sirolimus therapy while the mass slowly gets smaller. In the future, surgery might be necessary to remove the rest of the malformation.

## 3. Discussion

We share our experience of an airway-threatening cystic lymphatic malformation in a late preterm neonate that required a combination of various therapies. Prenatal diagnosis and assessment of the compression of surrounding tissues are vital for perinatal management. Ultrasound imaging combined with MRI allows for determining the compression of the trachea and the necessity for the EXIT procedure. The prenatal MRI could not be performed on our patient due to maternal COVID infection. Such examination is beneficial as it demonstrates the full extent of the mass and provides essential information for accurate treatment planning. Lesions in the frontal part of the neck create a higher risk for airway obstruction and adverse neonatal outcomes. Despite no success in the EXIT procedure, the baby’s airways were secured. A multidisciplinary approach was crucial as the treatment involved sclerotic agents, sirolimus therapy, and surgery.

The EXIT procedure is more and more often used to deliver infants with congenital head, neck, and intrathoracic masses and congenital diaphragmatic hernias after Fetoscopic Endoluminal Tracheal Occlusion and severe cardiac malformation [11,12]. The outcomes of EXIT itself are relatively good, and in the vast majority of cases, it enables establishing access to the airways [13]. In our case, the baby was intubated after clamping the cord as the EXIT procedure did not succeed, although it did not worsen the infant’s condition.

Injection of sclerotic agents (such as doxycycline, bleomycin, or OK-432) should be considered whenever the complete excision of cystic lymphatic malformation is impossible. Sclerotic agents are irritants that cause inflammation of the cyst lining, so the outflow of the lymphatic fluid can be diminished or even blocked. Sclerotherapy might have adverse effects such as bleeding, scarring, pulmonary fibrosis, and infection [14]. Doxycycline is widely used to treat postoperative lymphoceles and pleural effusion [15]. It is safe and effective in the pediatric population for managing lymphatic malformations [16]. In 2022 a Japanese group reported the successful application of eppikajyutsuto (TJ-28) in the sclerotization of lymphatic malformation of the newborn. This sclerotic agent has been known in Japanese traditional medicine for ages, but more studies are needed to assess its wider use [17].

Recently OK-432 has been more commonly used as it has even fewer adverse effects than doxycycline [18]. OK-432 is a lyophilized incubation mixture of group Streptococcus pyogenes of human origin that activates white blood cells and cytokines to increase endothelial permeability and accelerate the lymph drainage that leads to shrinkage of the cysts [19].

Sirolimus—the mammalian target of rapamycin (mTOR) inhibitor prevents angiogenesis and cellular proliferation. In the literature, off-label oral administration of this drug has been a successful treatment for lymphatic malformations in the pediatric and neonatal population [20]. The topical use of sirolimus has also been studied in randomized controlled trials in older children to treat microcystic lymphatic malformations [21,22].

Triana et al. reported seven cases of infants with a malformation located in a cervical region involving the upper airway. A satisfactory result was reached in all of them with an oral solution of sirolimus [23]. Changhua et al. reported eight patients who received treatment with sirolimus [24]. Like in the case we presented, only minor adverse effects were observed. The drug was efficient in seven patients, and one did not respond to the therapy; however, his clinical condition improved [24]. Azouz et al. reported a case study of a child affected by cystic lymphatic malformations from birth, delivered via EXIT procedure, who underwent multiple sclerotherapy sessions followed by sirolimus after the first month of age. The treatment continued for 15 months and resulted in a 90% decrease in the mass [25]. It is crucial to regularly and adequately monitor sirolimus concentration in plasma as cytochrome P450 enzymes metabolize it. Therefore, in neonates and infants, its concentrations may vary depending on the maturation of the liver. Generally, it is recommended to maintain lower plasma levels of sirolimus in neonates and infants than in older children, but more data on this drug’s pharmacokinetics are needed in neonates. In our patient, similarly to the literature data, no severe short-term toxicities of sirolimus have been observed. Sirolimus might help treat cystic lymphatic malformations, mainly composed of microcystic components, while macrocystic parts may require other therapies [25]. We speculate that there might be a synergic effect between the use of doxycycline, OK-432, and sirolimus.

In the literature, the case series might prove the success of various treatment strategies in children with head and neck lymphatic malformations. However, the patients differ substantially in terms of the malformation’s dimensions and response to the therapy [26,27]. According to the published data, we can speculate that some of the lesions become smaller after several months of treatment with sirolimus and thus may become feasible for surgical removal. In most cases, the neurodevelopmental milestones of the patients can be reached as long as their mobility is unaffected. The long-term outcome for adulthood remains unknown.

Our patient had a difficult course due to the location and size of lymphatic malformation, sepsis and possible inflammation of the cystic lesion, and respiratory problems (long need for mechanical ventilation). In such cases, a multidisciplinary team of experienced obstetricians, neonatologists, otolaryngologists, pediatric oncologists, and radiologists is required to provide optimal and state-of-the-art care. To achieve this, a special Section of Vascular Anomalies in Children was launched in 2021 under the auspices of the Polish Society of Pediatric Oncology and Hematology. It invites all experienced specialists dealing with all types of vascular lesions and malformations affecting children, including neonates, to provide the best possible multispecialty care over this heterogeneous, complicated, and demanding group of patients.

## Figures and Tables

**Figure 1 biomedicines-10-02422-f001:**
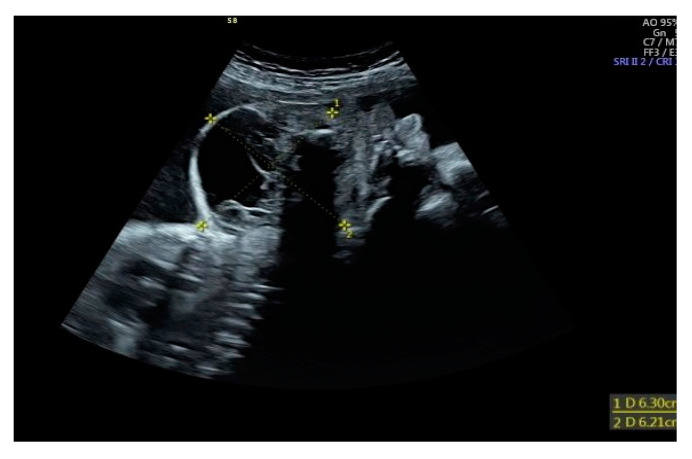
A longitudinal scan of the malformation on the 29th week of gestation. The lesion forced a retroflexed position of the fetal head.

**Figure 2 biomedicines-10-02422-f002:**
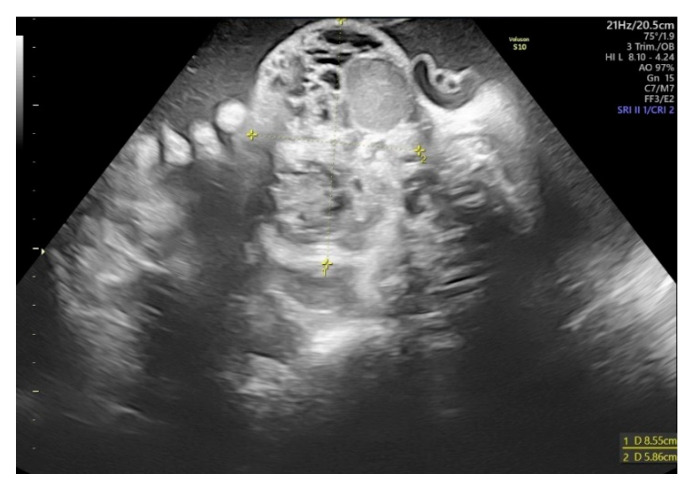
A transverse view of the fetal neck on malformation level on the 32nd week of gestation shows mixed echogenicity of the lesion with cystic and solid components.

**Figure 3 biomedicines-10-02422-f003:**
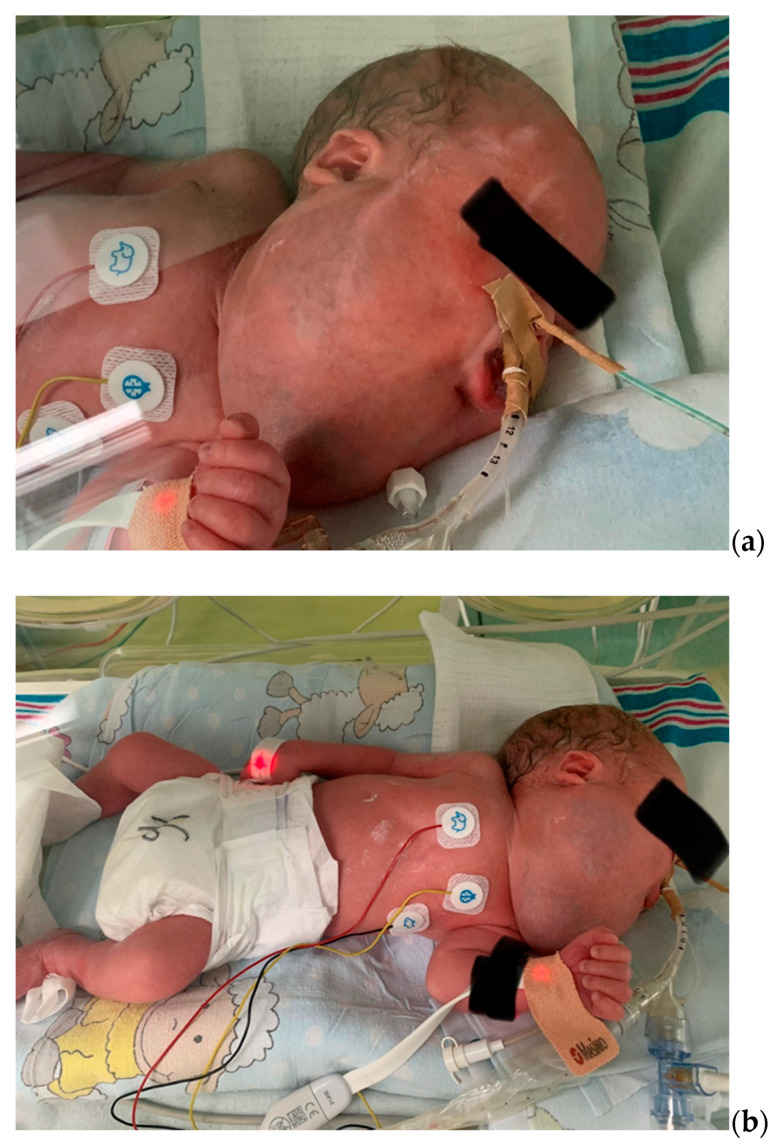
(**a**,**b**). The infant on the 1st day of life.

**Figure 4 biomedicines-10-02422-f004:**
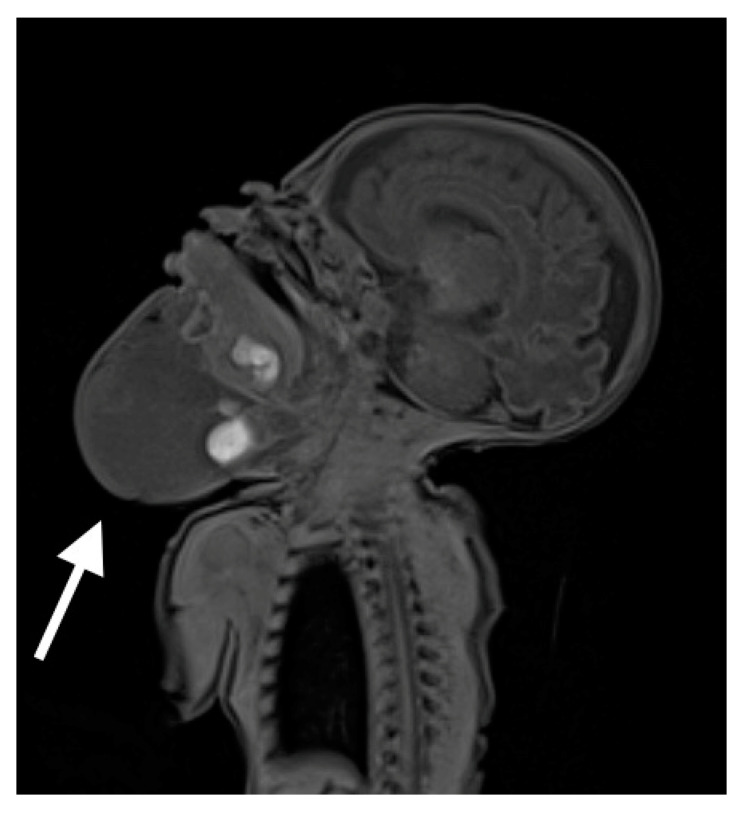
MRI scan on 6th day of life with visible cystic mass with approximate dimensions of cephalocaudal 10 cm × transverse 8 cm × anteroposterior 6.5 cm.

**Figure 5 biomedicines-10-02422-f005:**
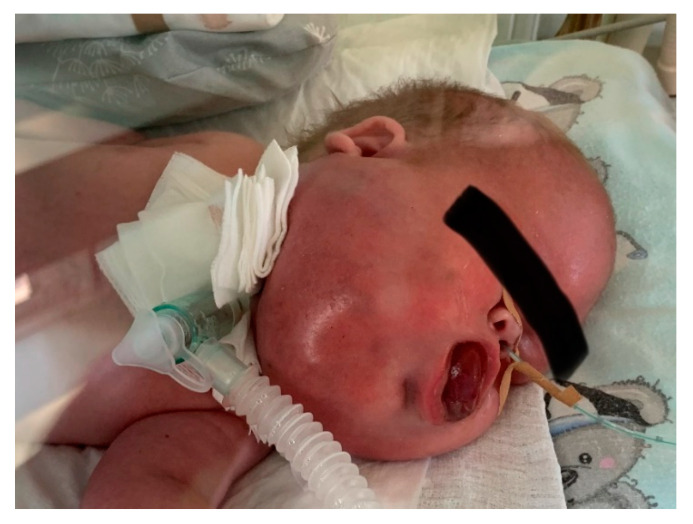
The infant with tracheostomy and lymphatic edema.

**Figure 6 biomedicines-10-02422-f006:**
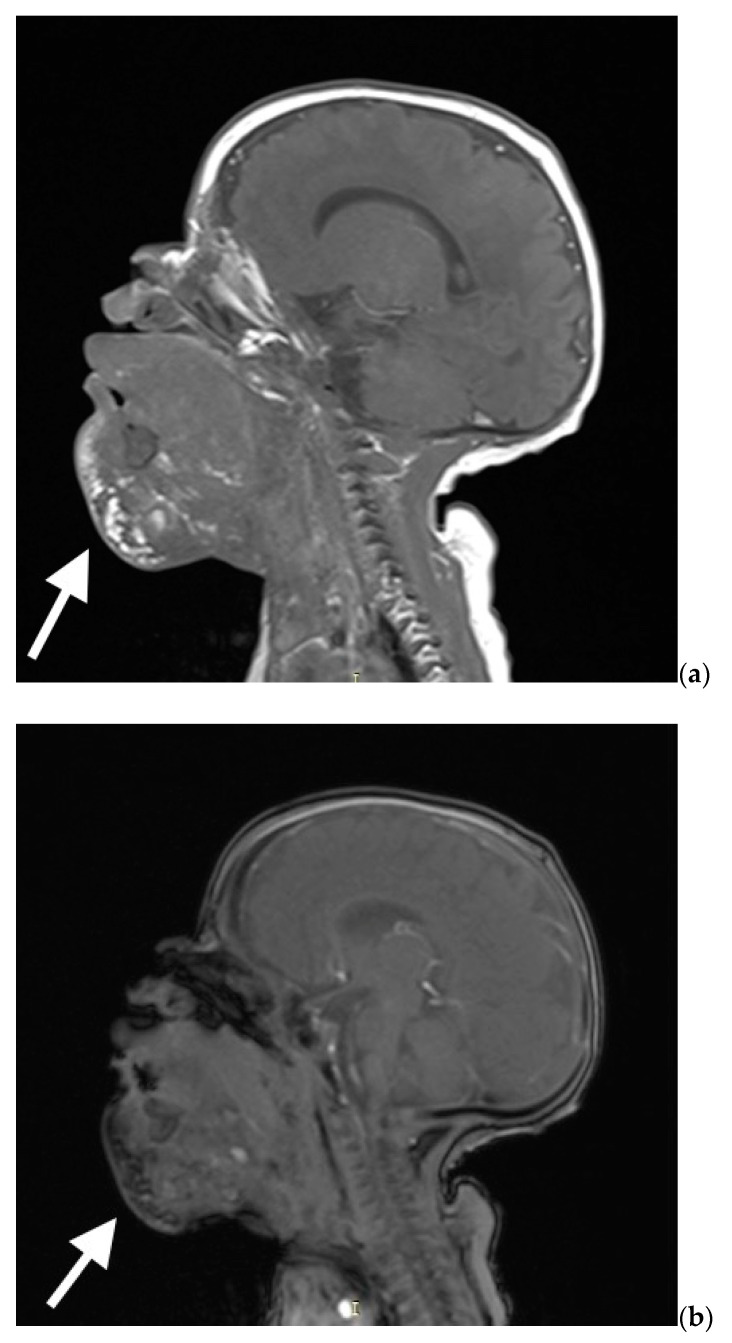
(**a**,**b**) MRI scan on 70th day of life—an extensive multicystic lesion with multiple septa penetrating both sides of the oral cavity, parotid salivary gland, and sublingual space, between the muscles of the oral cavity floor, including the whole submandibular area.

## Data Availability

Not applicable.

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
