# Peer review of "Multispecialty Approach to a Very Large Congenital Head and Neck Cystic Lymphatic Malformation in an Infant Born by SARS-CoV-2 Positive Mother—A Case Report"

_biomedicines, 2022, doi:10.3390/biomedicines10102422_

Round 1
Reviewer 1 Report
The authors described the multispecialty approach to a very large congenital head and neck cystic lymphatic malformation in an infant born by SARS-CoV-2 positive mother. The EXIT procedure was attempted.
The paper is well written. However, some issues remain.
Sometimes the authors used the term “tumor” to indicate the lymphatic malformation. However, it is not correct because such disease is not a neoplasia. Moreover, at line 113, the authors used the term “hemangioma”. Please correct the terms in the paper.
Please add the description of the lesion in the legends of figures 1 and 2
Why MRI had not been performed during gestation?
The authors should report the possible reasons for anemia, abnormal coagulation values, edema and hypoalbuminemia.
The description of the points were sclerotehrapy was performed must be added.
The partial resection must be better described. Which part of the malformation was removed?
White cells count must be added.
The treatments were not complete at the time of writing. I think that the paper should be resubmitted when all the treatments will be performed and the disease solved.
Author Response
非常感谢您关注我们在生物医学特刊上发表的手稿。感谢您阅读本文并提出许多有助于改进文章内容的重要建议。我们试图包括所有建议的修订。请在下面找到对您的评论的详细回复:
审稿人 1:
- As suggested, we changed the word “tumor” into “lesion” (lines 35, 58, 81, 225, 237), “malformation” (lines 60, 78, 91, 111, 115, 127, 144, 224) or “mass” (lines 62, 101). In some sentences, we left the word “mass” only instead of “tumor mass” (lines 117, 118, 128, 141, 161, 212)
We also changed the term “hemangioma” into “lymphangioma,” which is correct (line 114).
- We added the description of the lesions in the legend of figures 1 and 2.
Figure 1. A longitudinal scan of the lesion on the 29th week of gestation. The tumor forced a retroflexed position of the fetal head.
Figure 2. A transverse view of the fetal neck on lesion level on the 32nd week of gestation shows mixed echogenicity of the tumor with cystic and solid components.
- The MRI examination could not be performed before delivery as the mother suffered from COVID infection. We added this explanation in the Case description and the Discussion.
Lines 70-71: Due to this infection, we could not perform magnetic resonance imaging (MRI) before birth.
Lines 174-177: The prenatal MRI could not be performed on our patient due to maternal COVID infection. Such examination is beneficial as it demonstrates the full extent of the mass and provides essential information for accurate treatment planning.
- We added information about the possible reasons for anemia, abnormal coagulation values, edema, and hypoalbuminemia.
Lines 111-112: Anemia and abnormal coagulation values were probably due to consumptive coagulopathy and bleeding into the cysts.
Lines 114-116: Edemas were observed in the lesion area due to malfunctioning lymphatic vessels and very limited motor activity of the baby. Hypoalbuminemia was most probably reactive to edema.
- We added the description of the points where sclerotherapy was performed.
Lines 154-156: During this procedure, similarly to doxycycline injection, the obliterating preparation was administered to three malformation chambers with a diameter exceeding 20 mm.
- We added more information on the partial resection of the malformation.
Lines 127-128: The tumor parts limiting access to the anterior wall of the trachea were removed.
- We added the information on white blood cell count.
Line 144: …leucopenia with the lowest white blood cell count 2.42 G/L.
- 很难说婴儿的治疗何时结束。大概是3到5年的前景。然而,治疗中最重要的部分已经完成。目前,婴儿在家中情况稳定,仅接受西罗莫司。这种治疗可能会持续几年。这就是我们决定现在就撰写并提交手稿的原因。
我们更改了病例报告中的最后一句话,更新了患者的信息:
第 174-176 行。目前,婴儿 7 个月大,仍在接受西罗莫司治疗,而肿块逐渐减少。将来,可能需要手术切除其余的畸形。

Reviewer 2 Report
Thank you for the privilege of reviewing this case report. I ask authors to perform some modifications.
line 21: replace "Clinical" by "We analyzed clinical".
Line 22: delete: "were performed"
Line 37: replace "stomach" by "gastric tumors".
Line 41: replace: "those" by "these".
Line 134: replace "standard" by "stable".
Line 198: replace: "successive" by "successfull"
line 215: replace "no short-term severe" by "no severe short-term".
Author Response
We changed all the expressions as required.
Round 2
Reviewer 1 Report
The term "tumor" is still used in some cases in the manuscript. Please correct all the terms.
Author Response
We replaced or removed the word "tumor" in the lines where it was left:
lines 86 and 90 - replaced with "lesion"
line 129 - removed
The only place where the word "tumor" is left is the line 37 but it applies to other organs and was required by the second Reviewer.